# Brain Redox Imaging Using In Vivo Electron Paramagnetic Resonance Imaging and Nitroxide Imaging Probes

**Hirotada G. Fujii [1],\*, Miho C. Emoto [2] and Hideo Sato-Akaba [3]**

[1] Health Sciences University of Hokkaido, Ishikari, Hokkaido 061-0293, Japan
[2] Health Sciences University of Hokkaido, Sapporo, Hokkaido 002-8072, Japan; emoto@hoku-iryo-u.ac.jp
[3] Department of Systems Innovation, Graduate School of Engineering Science, Osaka University, Osaka 560-8531, Japan; akaba@sup.ee.es.osaka-u.ac.jp
\* Correspondence: hgfujii@hoku-iryo-u.ac.jp; Tel.: +81-90-5072-6192

**Abstract:** Reactive oxygen species (ROS) are produced by living organisms as a result of normal cellular metabolism. Under normal physiological conditions, oxidative damage is prevented by the regulation of ROS by the antioxidant network. However, increased ROS and decreased antioxidant defense may contribute to many brain disorders, such as stroke, Parkinson's disease, and Alzheimer's disease. Noninvasive assessment of brain redox status is necessary for monitoring the disease state and the oxidative damage. Continuous-wave electron paramagnetic resonance (CW-EPR) imaging using redox-sensitive imaging probes, such as nitroxides, is a powerful method for visualizing the redox status modulated by oxidative stress in vivo. For conventional CW-EPR imaging, however, poor signal-to-noise ratio, low acquisition efficiency, and lack of anatomic visualization limit its ability to achieve three-dimensional redox mapping of small rodent brains. In this review, we discuss the instrumentation and coregistration of EPR images to anatomical images and appropriate nitroxide imaging probes, all of which are needed for a sophisticated in vivo EPR imager for all rodents. Using new EPR imaging systems, site-specific distribution and kinetics of nitroxide imaging probes in rodent brains can be obtained more accurately, compared to previous EPR imaging systems. We also describe the redox imaging studies of animal models of brain disease using newly developed EPR imaging.

**Keywords:** ROS; oxidative stress; redox status; EPR imaging; MRI; brain disease; antioxidant

## 1. Introduction

Reactive oxygen species (ROS) are recognized to play important roles in both physiological and pathological processes [1,2]. ROS are produced by living organisms as a result of normal cellular metabolism. Under normal physiological conditions, oxidative damage is prevented by the regulation of ROS through the antioxidant effects of a network of enzymatic and nonenzymatic systems. However, increases in concentrations of ROS and decreases in the antioxidant defense systems may contribute to aging, cancer, heart failure, and brain disorders such as stroke, Parkinson's disease, and Alzheimer's disease (AD) [3,4], as a result of a loss of balance in the redox status due to dysfunctions in such defense systems caused by ROS. Noninvasive assessment of the redox status in the body is thus extremely important for monitoring the disease state and also clarifying the role of ROS in the development and causes of such diseases.

Electron paramagnetic resonance (EPR) is a spectroscopic method of measuring ROS and many other free radicals directly or indirectly with the aid of chemical compounds, as a so-called "spin trap" [5,6]. Since levels of bioradicals including ROS and reactive nitrogen species (RNS)

generated in living organisms are very low, in vivo direct or indirect detection of bioradicals by EPR seems quite impossible. On the other hand, in vivo EPR imaging using nitroxides as imaging probes has been emerging since the early 1980s as a powerful tool to visualize free-radical reactions and the distribution of nitroxide imaging probes in biological specimens [7–12].

Nitroxides are stable organic free-radical compounds with a single unpaired electron and provide T1 contrast enhancement in magnetic resonance imaging (MRI) [13,14]. The only chemical drawback of nitroxides is their susceptibility to reduction to the corresponding diamagnetic hydroxylamine, resulting in loss of paramagnetism [15,16]. However, their properties allowing them to undergo bioreduction can provide useful information pertaining to biochemical reactions in organisms. Nitroxides are redox-active species that can be oxidized or reduced by the corresponding chemical reactants in cells, and participate in cellular redox reactions. The reduction reaction rate of nitroxides may thus offer an index of global cellular redox status. As a result, monitoring the rate of transformation of nitroxides to the corresponding diamagnetic species by EPR imaging can provide an in vivo assessment of redox status in animal disease models. Such redox mapping based on redox-sensitive paramagnetic spin probes has been carried out by EPR imaging. For conventional continuous-wave EPR imaging, however, poor signal-to-noise ratio, low acquisition efficiency, and a lack of anatomic visualization limit the ability to produce three-dimensional (3D) redox maps of small rodent brains. Moreover, nitroxide-based imaging probes have not been well described in terms of the distributions and kinetics in vivo, due to a lack of adequate EPR imagers. This review discusses the instrumentation and coregistration of EPR images to anatomical images and appropriate nitroxide imaging probes, all of which are needed for sophisticated in vivo EPR imaging of small rodents. We also described redox imaging studies of animal models of brain disease using the newly developed EPR imaging system.

## 2. In Vivo EPR Imaging Instrument for Small Animals

EPR spectroscopy and imaging can be used to investigate chemical reactions that involve free radicals. Visualizing the dynamics of such chemical reactions within a reasonable timeframe requires a relatively fast data acquisition and imaging method. As with nuclear magnetic resonance (NMR), pulsed EPR spectroscopy was developed to rapidly measure the EPR spectra of bioradicals. However, recent pulsed EPR methods have remained limited to nitroxides with relatively long relaxation times [17]. In contrast, the continuous-wave EPR (CW-EPR) method can be applied to most nitroxide imaging probes regardless of their relaxation time. For general CW-EPR measurements, relatively slow scanning of magnetic fields is used to measure the hyperfine coupling structures, and the ideas of rapid scanning and accumulation thus have not been considered to be appropriate previously in EPR imaging experiments.

Temporal changes in EPR signal intensity for nitroxide imaging probes in the mouse head have been observed after injection through the tail vein. The EPR signal intensity of injected nitroxide measured in the mouse brain peaked at around 20 s after injection, and intensity gradually decreased depending on the redox status of the examined mouse [18,19]. Therefore, to visualize the distribution of nitroxide probes in the mouse brain at the time of maximal signal intensity, acquisition of spectral data for reconstruction of the EPR image of the mouse head needs to be as rapid as possible. Total acquisition time for EPR imaging is a product of the number of projections (number of spectra under magnetic field gradient) and the acquisition time of an EPR spectrum. Thus, to achieve an image from 100 projection data in less than 10 s, one spectrum should be measured and recorded in 100 ms. One possible method to reduce the imaging time using CW-EPR is to introduce rapid magnetic field scanning.

In our first in vivo EPR imaging experiment for mouse cancer, total acquisition time was 8 min (2-min scanning time × 4 projections) [8]. After this study, the goal of reducing imaging time was started while fighting to reduce distortion of the recorded spectra. In 1988, Demsar et al. [20] reported a scanning time of 6 s for X-band EPR imaging, and in 1990, Alecci et al. [21] reported 3.5 s as the scanning rate. In 1996, Yokoyama et al. [22] achieved a scanning rate of 1.4 s with an air-core Helmholtz coil pair, and in 2007, Samouilov et al. [23] reported in vivo EPR imaging of mice with

a scanning rate of 1.3 s. In the beginning of the 2010s, Hirata H. from Hokkaido University started developing an in vivo EPR imaging system with rapid field scanning capability. Using the developed EPR imager, Sato-Akaba et al. [24] were able to take 3D EPR images of a phantom with an interval of 3.6 s, and Fujii et al. [25] succeeded in obtaining a series of 3D EPR images of mouse heads every minute continuously.

Figure 1A shows a schematic of a 750-MHz CW-EPR imager [26]. A magnetic circuit of 27 mT and a multicoil parallel-gap resonator were used with the developed CW-EPR digital console. To reduce total acquisition time, analog signals were used to drive the Helmholtz coil pair for field scanning and field gradient coils were used. These signals were generated simultaneously with four 12-bit digital analog converters controlled by a field programmable gate array (FPGA) developing board (DE0-nano) in the sequence controller. Figure 1B shows the timing chart and control signals for the field scanning, magnetic field gradients, and data acquisition in the CW-EPR imager.

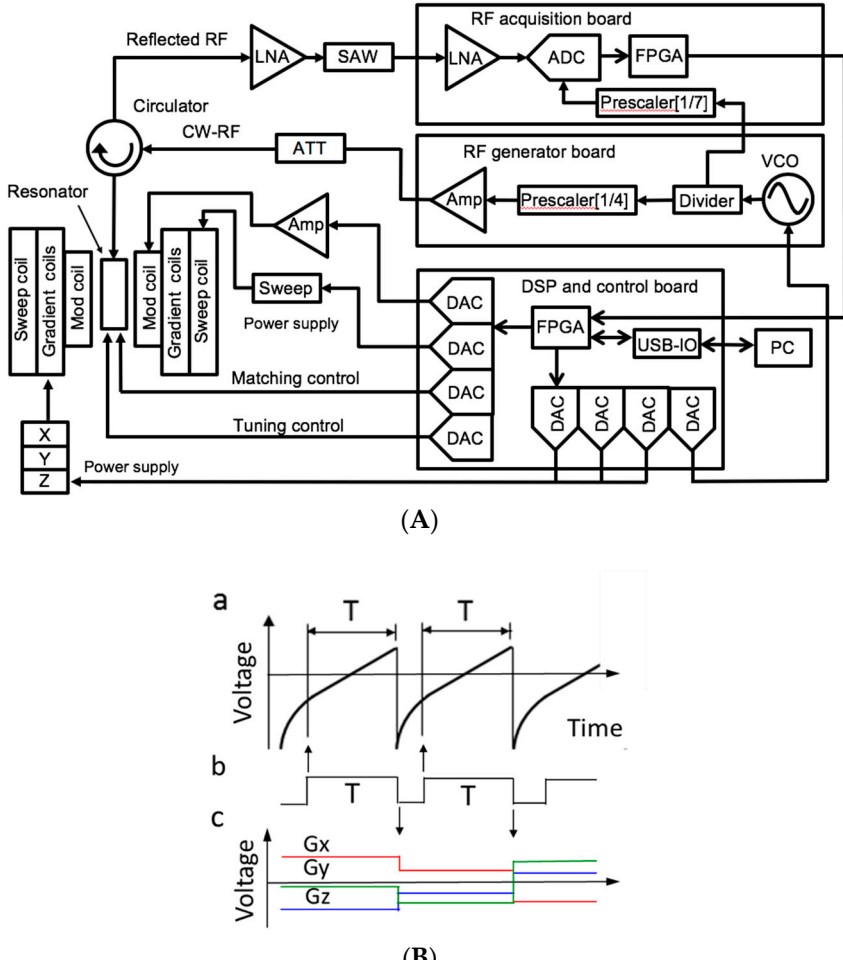

**Figure 1.** (**A**) Schematic of the in vivo digital EPR imager [26]; (**B**) timing chart and signals for the (**a**) field scan, (**b**) trigger pulse used for the data acquisition window with a period of T, and (**c**) 3D field gradients Gx, Gy, and Gz.

## 3. Nitroxide Imaging Probes for EPR Imaging Study

Low-molecular-weight, stable nitroxide free radicals have been used as reporter probes in spin-labeling biochemistry and biophysics, with an X-band EPR spectrometer being employed to detect such information. Many nitroxide compounds with different chemical structures have also been used as imaging probes for in vivo EPR imaging studies (Figure 2). Piperidine nitroxides such as Tempol (A) have been widely used for in vivo EPR studies. The first report of in vivo EPR

imaging of mouse cancers introduced 3-carboxamido-2,25,5-tetramethylpyrrolidine-1-oxyl (CTPO, (B) in Figure 2) [8] imaging probes, since the in vivo lifetimes of these probes are relatively longer than those of six-membered piperidine nitroxides. Since that study, five-membered pyrrolidine nitroxides ((C) and (D) in Figure 2) have gained wide use in most EPR imaging studies due to their long lifetimes in biological systems [10,27,28].

**Figure 2.** Nitroxide imaging probes used for EPR imaging: (**A**) Tempol, (**B**) CTPO, (**C**) CMP, (**D**) COP, (**E**) HMP, and (**F**) MCP.

The topic of free-radical reactions in the brain is one of the most important subjects in recent medical research. For brain redox imaging by an EPR imager, nitroxides that can pass through the blood–brain barrier (BBB), are retained within the brain, and are sensitive to redox status under oxidative stress are desirable. Sano et al. [29] synthesized a BBB-permeable nitroxide imaging probe, 3-methoxycarbonyl-2,2,5,5-tetramethyl-pyrrolidine-1-oxyl (MCP, Figure 2F), and the distribution of MCP in mouse brains was visualized by 2D EPR imaging. Yokoyama et al. [30] also synthesized 3-hydroxymethyl-2,2,5,5-tetramethylpyrrolidine-1-oxyl (HMP, Figure 2E) and MCP, and examined the distributions of these two probes in rat brains using EPR imaging. Interestingly, EPR images using these probes showed regional differences in decay rates in the rat brain.

3D EPR images of mouse heads using BBB-permeable MCP and BBB-impermeable COP were taken (Figure 3). The EPR images showed that MCP is distributed mainly inside the brain, but COP is distributed mainly outside the brain. Partition coefficients of MCP and COP between n-octanol and water were used as indices of the lipophilicity of these nitroxides, with the values suggesting the different distributions of these probes in mouse heads [25,30].

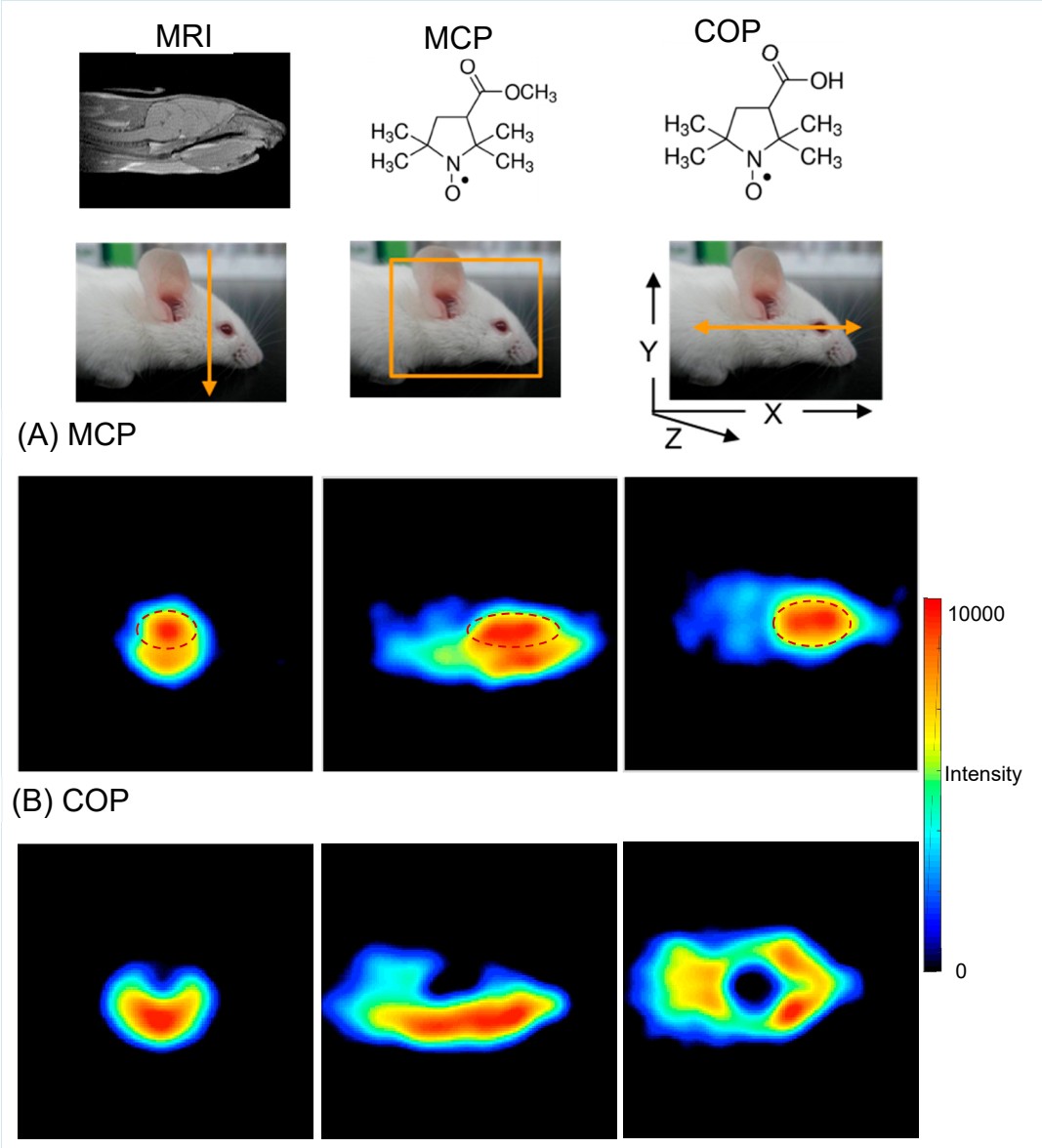

**Figure 3.** Distribution of blood–brain barrier (BBB)-permeable nitroxide, MCP, and BBB-impermeable nitroxide, COP, in mouse heads. (**A**) 2D slice images of MCP distribution. The area of the brain tissue is indicated by the dotted line. (**B**) 2D slice images of COP distribution in mouse heads. BBB-impermeable COP was distributed outside the mouse brains. 2D slice images (128 × 128 pixels) were generated from the 3D image data (128 × 128 × 128 pixels) [24,25].

The spatial resolution of EPR images depends on the line-widths of nitroxide imaging probes and the magnetic field gradients. With a process of deconvolution for recovering the spatial distribution of EPR signals, spatial resolution of less than 1 mm was possible in the improved EPR imaging system [25,26].

One of the interesting properties of nitroxides is the ability to distinguish the EPR spectra of two nitroxide probes by incorporating the $^{14}$N and $^{15}$N nitrogen isotopes. These $^{14}$N- and $^{15}$N-labelled nitroxide probes show three-line and two-line EPR spectra, respectively, and each EPR spectrum is clearly separated under no magnetic field gradient even after simultaneous administration of both probes [31]. With the administration of BBB-impermeable $^{14}$N-labeled COP and BBB-permeable $^{15}$N-labeled MCP into mice, the EPR image clearly shows distributions of MCP and COP in the same mouse simultaneously. For simultaneous molecular imaging studies by EPR, the synthesis of new

isotope-labeled probes [32,33] and the development of a simultaneous imaging method [31,34] to visualize target molecules have been carried out.

## 4. Coregistration EPR/NMR Imaging

EPR imaging can enable spatial mapping of free-radical-related information in biological specimens and offers a unique functional imaging method. The distribution of most nitroxide imaging probes is not identical to that of water molecules in vivo, especially for biological objects. Therefore, finding and indicating the exact location of free-radical information in biological objects requires coimaging of the obtained EPR image with an anatomical image taken using some other imaging modality. For visualization of water molecules with high temporal and spatial resolution in vivo, in general, proton NMR (MRI) is well suited to providing anatomical visualization and is widely used for coimaging with other modalities. At present, MRI is the major imaging modality to have been used in coimaging with EPR imaging.

Two types of coimaging methods are available. The first approach is to build EPR/MRI coimaging instrumentation. Zweier's group reported several types of EPR/MRI coimaging system, enabling coimaging of the whole body of mice without sample movement and visualization of organ-specific redox and oximetry information. These coimaging EPR/MRI systems use a dual-frequency resonator for EPR imaging and MRI [23,35,36]. Fujii et al. [37] developed an EPR/MRI dual-imaging system with one permanent magnet, in which one part is adjusted to 0.5 T for MRI and the other to 0.042 T for EPR (1.2 GHz). Samples can be shuttled between MRI and EPR resonators. The second approach for coimaging is to transfer a subject animal between EPR and MRI instruments in a specially designed animal holder. Position markers are also used to adjust the registration of both images [23,35]. Adjustment of both images using position markers is simple, and the accuracy of the image registration depends largely on the spatial resolution of EPR images.

## 5. Brain Redox Imaging by CW-EPR Imager

Nitroxides are valuable biophysical probes that have been used in both in vitro and in vivo EPR studies. Nitroxides have recently been identified to act as biological antioxidants [38,39] and radio-protectors [40] in biological systems. EPR imaging using nitroxide imaging probes is therefore a unique functional imaging modality.

Nitroxides exist in biological systems as a redox pair; i.e., the nitroxide free-radical form (EPR-active form) and the diamagnetic hydroxylamine form (EPR-silent form), which is the one electron reduction product of the nitroxide free radical.

Nitroxides are redox-active species and participate in cellular redox reactions, and many studies have indicated that the levels of nitroxides in biological systems are indicative of the global cellular redox status. The conversion of nitroxide to diamagnetic species is significantly retarded by lower levels of thiol compounds such as glutathione, one of the major antioxidants. The redox environment within living cells is thus an important parameter that may indicate levels of oxidative stress, disease state, aging state, and more [15,41,42]. Given this background, noninvasive visualization of the redox status by EPR imaging using nitroxide imaging probes is expected to help provide versatile medical information, especially for brain diseases such as AD and Parkinson's disease.

### 5.1. Three-Dimensional EPR Imaging of the Mouse Head

As shown above, to visualize the distribution of nitroxide imaging probes in living organisms, the acquisition time for a dataset of spectral projections should be shorter than the lifetime of the nitroxide probes. Therefore, to visualize the 3D-distribution of nitroxide probes in small rodents, faster EPR imagers are absolutely required. The first EPR image of mouse heads using BBB-permeable 4-hydroxy-2,2,6,6-tetramethylpperidine-d17-1-15N-1-oxyl was successfully obtained by Sato-Akaba et al. [24], in which a set of projection data for 3D EPR imaging was acquired every 30 s with field scanning of 0.5 s and the number of projections set at 46. Their method enables a change in the number of averagings for measured spectra in post-processing of image reconstruction, since several datasets can be recorded during the time in which EPR spectra of nitroxide probes are detectable. With further improvement of a fast EPR imager, Fujii et al. obtained 3D EPR images of mouse brains using the BBB-permeable nitroxides HMP and MCP and BBB-impermeable COP [26]. BBB-impermeable COP cannot enter the brain tissue of healthy mice, but COP can enter the brain tissue through wound sites or breaks in the BBB. This phenomenon was visualized by EPR imaging of the infarcted hemisphere in a mouse model of ischemia–reperfusion, where COP was distributed in the infarcted hemisphere, but not in the intact hemisphere [43].

### 5.2. Redox Mapping

The series of EPR images can be used to follow time-dependent changes in each voxel within the image (Figure 4A). The rate constants of the reduction reaction of the nitroxide probe for each pixel of the 2D temporal EPR images were calculated, and the 2D spatial mapping of pseudo-first-order rate constants of nitroxide imaging probes were presented in "redox mapping" of a subject sample. The use of redox mapping enables not only changes in the redox status of a subject to be followed, but also identification of organ-specific characteristics of the redox balance [44–46].

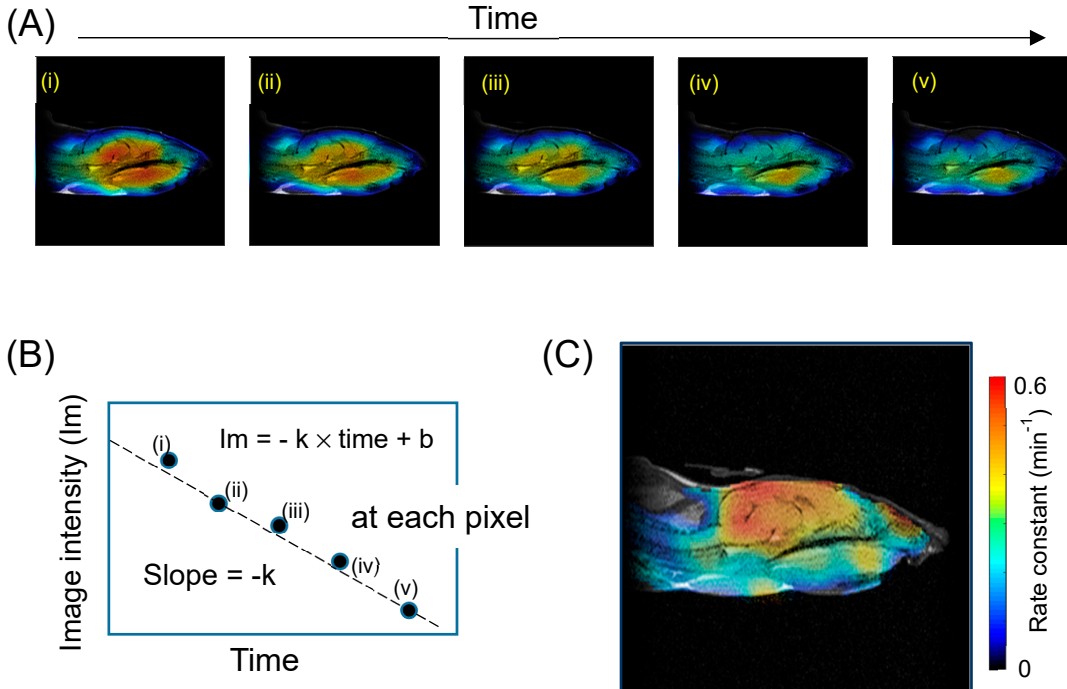

**Figure 4.** Redox mapping of mouse heads. (**A**) The 2D slice EPR images were obtained from the reconstructed 3D data of mouse heads. (**B**) The time course of the image intensity (Im) at each pixel was calculated ((**i**)–(**v**)), k: pseudo-first order rate constant. (**C**) 2D spatial mapping of the reduction rate constants of MCP was shown as redox mapping. Adapted from [19].

Kuppusamy et al. introduced "redox mapping" by EPR imaging to visualize tumor redox status [47], and redox mapping has since been widely used in many other examples. Brain redox imaging was first demonstrated in a mouse model of ischemia–reperfusion by 3D EPR imaging, and the obtained multislice redox mapping visualized a heterogeneous distribution of the reduction rate constant of nitroxides in the infarcted hemisphere of examined mouse heads [25]. After this work, brain redox mapping was widely used in brain disease models, such as psychostimulant methamphetamine-treated mice [48], septic mouse brain induced by lipopolysaccharide [49], and the pentylenetetrazol-induced kindling model of epilepsy [18]. With the increasing detection sensitivity of EPR imagers, region-specific analysis of redox imaging data has become possible. To obtain site-specific redox information, coregistration of the anatomical image to the redox information is necessary, especially for medical EPR imaging studies. In a transgenic mouse model of AD, redox data from the cortex, midbrain, and hippocampus were analyzed using coregistration of the redox map obtained by EPR and the anatomical image by MRI (Figure 5). This study clearly showed that 3D EPR imaging can detect the accelerated change in redox status of the AD mouse brain compared with control, and in particular can visualize a significant change in redox status in the hippocampus of the AD mouse brain [50].

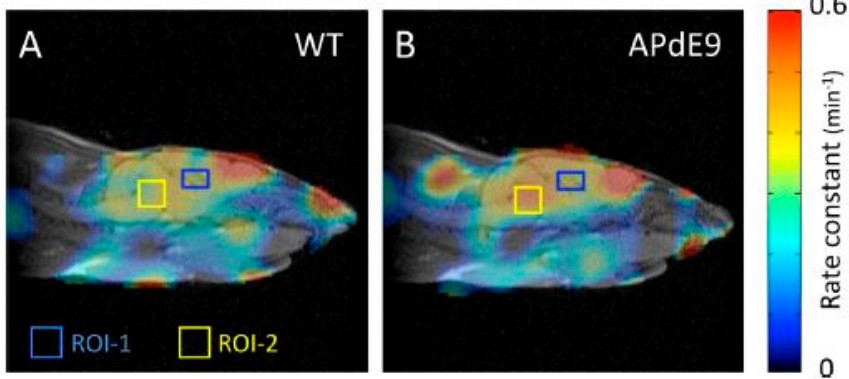

**Figure 5.** Coregistration of the redox map and the anatomical magnetic resonance imaging (MRI). The 2D spatial mapping of pseudo-first-order rate constants of nitroxide imaging probes were presented in "redox mapping" of examined mouse heads. Redox maps of a wild type (WT, (**A**)) and AD (APdE9, (**B**)) model mice. ROI-1: ROI in the hippocampus; ROI-2: ROI in the midbrain. Adapted from [50].

Conventional EPR imaging systems did not show structure-specific redox information in rodent brains. However, with new EPR imaging systems, detailed information of nitroxide distribution and kinetics within rodent brains can be obtained more accurately.

*5.3. Glutathione (GSH) Mapping*

GSH is an important antioxidant that can protect cells under oxidative stress. The maintenance of GSH levels is important to prevent neuronal disorders such as AD and Parkinson's disease [51,52]. The reduction reaction of nitroxides in vivo depends on the concentration of ascorbic acid, and this reaction is catalyzed, depending on GSH levels in vivo [19,53]. Using this property, a new method to image the distribution of GSH levels in specific brain regions was developed and successfully applied to a kindling mouse model. The obtained map of brain GSH levels clearly visualized decreased GSH levels around the hippocampus region.

**6. Conclusions**

To visualize brain redox status in small rodents, in vivo EPR imaging instruments with fast data acquisition capability and higher detection sensitivity are desirable. High-resolution EPR imagers can provide site-specific redox information, so coregistration of EPR imaging to anatomical maps is

needed for medical imaging studies. Other imaging modalities such as MRI should be used with EPR imagers. Lastly, development of novel nitroxide imaging probes is needed for medical imaging studies [54]. BBB-permeable, nontoxic nitroxide imaging probes are necessary for brain redox studies. Rapidly reduced nitroxide probes show higher redox sensitivity, so EPR imagers with rapid data acquisition capabilities are appropriate. With the development of EPR imagers with the capability for fast data acquisition, BBB-permeable nitroxide probes, and anatomical imaging tools such as MRI in the laboratory, EPR imaging systems are becoming a powerful imaging tool that can contribute in important ways to elucidation of the oxidative diseases of the brain.

**Funding:** This research received no external funding.

**Acknowledgments:** Our work was supported in part by JSPS KAKENHI Grant Number 26293280 (HGF) and 16K10291 (MCE).

**Conflicts of Interest:** The authors declare no conflict of interest.

## Abbreviations

CMP      3-carbamoyl-2,2,5,5-tetramethylpyrrolidin-1-oxyl
COP      3-carboxy-2,2,5,5-tetramethyl-1-pyrrolidin-1-oxyl
CTPO     3-carbamoyl-2,2,5,5-tetramethyl-3-pyrrolin-1-oxyl
MCP      3-methoxycarbonyl-2,2,5,5-tetramethylpyrrrolidine-1-oxyl
Tempol   4-hydroxy-2,2,6,6-tetramethylpiperidine-1-oxyl
HMP      3-hydroxymethyl-2,2,5,5-tetramethylpyrrolidine-1-oxyl
ADC      analog-to-digital converter
AFC      automatic frequency control
Amp      amplifier
BPF      band pass filter
DAC      digital analog convertor
DDC      digital down-converter (DDC)
DSP      digital signal processor
FPGA     field programmable gate array
LNA      a low noise amplifier (LNA)
RF       radio frequency
VCO      voltage controlled oscillator

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
