# Peer review of "Brain Redox Imaging Using In Vivo Electron Paramagnetic Resonance Imaging and Nitroxide Imaging Probes"

_magnetochemistry, doi:10.3390/magnetochemistry5010011_

Round 1

Reviewer 1 Report

I think this paper is fine for publication in Magnetochemistry.   I am not qualified to credibly comment on the imager design. My main issue with this "redox" mapping strategy is the lack of negative control - are the pharmacokinetics of the nitride radicals the same in the AD and normal mouse brains... signal loss could also be due to rapid washout of nitrides.. not just reduction. I do acknowledge that the control experiment will not be easy, the authors would probably need to track radiolabelled nitroxide in both normal and AD brain - well beyond the scope of this paper.

One mistake: the scale bar in Fig 5 is not "rate constant", it should be "observed rate."

Otherwise paper is fine to publish in Magnetochemistry in my opinion. 

Author Response

Reviewer-1 (R1)

Thank you very much for reviewing our manuscript and your interest in our research. Regarding your main issue, as you wrote, it was beyond the scope of this review, but I will answer to your question. As is shown in the research papers, EPR signal loss in the brain is mainly due to reduction of the nitroxide imaging probe, in case of MCP. The pharmacokinetics of the nitroxides (MCP) for control and AD mice did not differ until deposition of amyloid b protein in mouse brain.

R1-comment_1:  One mistake: the scale bar in Fig 5 is not "rate constant", it should be "observed rate."

[Response] Fig 5 is the two-dimensional spatial mapping of pseudo-first order rate constants. Thus, the scale bar is “Rate constant (min-1)”.

To avoid reader’s misunderstanding, the word “the pseudo-first order reaction rate” is added in the Figure legend of Fig. 5 in the revised manuscript.

Reviewer 2 Report

The paper gives a nice overview on brain imaging application, methods and spin probes used in the past and now. Interesting differentiation between BBB permeable and impermeable nitroxides. The paper is well written and structured.

Minor remarks:

In Fig 4 the individual images in B/C are not visible. Show them or leave them out.

What is the spatial resolution of the different EPR methods presented, could you provide values?

The papers below could be considered:

 In vivo evaluation of different alterations of redox status by studying pharmacokinetics of nitroxides using magnetic resonance techniques.

Bačić G, Pavićević A, Peyrot F.

Redox Biol. 2016 Aug;8:226-42. doi: 10.1016/j.redox.2015.10.007. Epub 2015 Nov 14. Review.

Imaging thiol redox status in murine tumors in vivo with rapid-scan electron paramagnetic resonance.

Epel B, Sundramoorthy SV, Krzykawska-Serda M, Maggio MC, Tseytlin M, Eaton GR, Eaton SS, Rosen GM, Kao JPY, Halpern HJ.

J Magn Reson. 2017 Mar;276:31-36. doi: 10.1016/j.jmr.2016.12.015. Epub 2016 Dec 31.

Author Response

Reviewer-2 (R2)

Thank you very much for reviewing our manuscript and your interest in our research.

R2-comment_1:  In Fig 4 the individual images in B/C are not visible. Show them or leave them out.

[Response]   According to the reviewer’s suggestion, the overlapped images in Fig. 4B were deleted. The letter “C” in Fig. 4C was invisible in the original manuscript. Thus, in the revised manuscript the letter was clearly illustrated in the black.

R2-comment_2:  What is the spatial resolution of the different EPR methods presented, could you provide values?

[Response] The following sentences were added in the revised manuscript.

“The spatial resolution of EPR images depends on the line-widths of nitroxide imaging probes and the magnetic field gradients. With a process of deconvolution for recovering the spatial distribution of EPR signals, the spatial resolution less than 1 mm was possible in the improved EPR imaging system.” 

R2-comment_3:  The papers below could be considered.

[Response] According to the reviewer’s suggestion, two papers were included in the reference lists of the revised manuscript.

1.           Bacic, G.; Pavicevic, A.; Peyrot, F. In vivo evaluation of different alterations of redox status by studying pharmacokinetics of nitroxides using magnetic resonance techniques. Redox Biol 2016, 8, 226-242, doi:10.1016/j.redox.2015.10.007.

2.           Epel, B.; Sundramoorthy, S.V.; Krzykawska-Serda, M.; Maggio, M.C.; Tseytlin, M.; Eaton, G.R.; Eaton, S.S.; Rosen, G.M.; Kao, J.P.Y.; Halpern, H.J. Imaging thiol redox status in murine tumors in vivo with rapid-scan electron paramagnetic resonance. J Magn Reson 2017, 276, 31-36, doi:10.1016/j.jmr.2016.12.015.

Reviewer 3 Report

In the current manuscript entitled "Brain redox imaging using in vivo electron paramagnetic resonance imaging and a nitroxide imaging probe,” the authors review the recent literature, advances and novelty of in vivo EPR imaging of the brain as a methodology to study brain redox status. The authors were able to describe a) advances and limitations of past and current instrumentation; b) pros and cons of spin probes to measures redox status via EPR imaging; and c) highlight the need for co-registration of EPR and MRI to obtain reliable redox imaging data. This review is well-written, but there are some minor grammatical errors that should be evaluated and corrected, and several minor comments should be addressed. The authors conclude that EPR imaging coupled with fast data acquisition, sufficient BBB-permeable spin probes and MRI co-registration may become a power tool to asses oxidation in the brain.

Minor Comments:

1.    The authors should define all abbreviations in the manuscript, particularly abbreviations for the nitroxides presented and discussed.

2.    The authors should show all the images in Fig 4 on the Time-axis, and is the y-scale in Fig 5 a rate constant measurement?

3.    Additionally, what is the scale for the colored bar on Fig 3? Intensity? The area where the brain in located should also be highlighted in the figure or figure legend where the brain is anticipated to be located to further emphasize and differentiate the differences in the distribution of both nitroxides. This may not be apparent to readers.

4.    The authors comment that nitroxide probes distribution and kinetics in vivo has not been well described. While this may be true for several of the nitroxides discussed in the manuscript, a review of the literature will reveals several manuscripts which discuss these parameters, particularly with reference to EPR imaging applications, of very similar nitroxides in vivo, in the brain.

5.    The authors should describe or distinguished what is meant by 3D EPR imaging compared 2D-spatial mapping. Are these interchangeable or are the key differences in methodology.

6.    Lines 142-144 are confusing as written.

Author Response

Reviewer-3 (R3)

Thank you very much for reviewing our manuscript and your interest in our research.

R3-comment_1:  The authors should define all abbreviations in the manuscript, particularly abbreviations for the nitroxides presented and discussed.

[Response]   According to reviewer’s suggestion, abbreviations for nitroxides were listed in the revised manuscript.

CMP:  3-Carbamoyl-2,2,5,5-tetramethylpyrrolidin-1-oxyl

COP:  3-Carboxy-2,2,5,5-tetramethyl-1-pyrrolidin-1-oxyl

CTPO:  3-Carbamoyl-2,2,5,5-tetramethyl-3-pyrrolin-1-oxyl

MCP:   3-methoxycarbonyl-2,2,5,5-tetramethylpyrrrolidine-1-oxyl

Tempol:  4-Hydroxy-2,2,6,6-tetramethylpiperidine-1-oxyl

HMP:   3-hydroxymethyl-2,2,5,5-tetramethylpyrrolidine-1-oxyl

R3-comment_2:  The authors should show all the images in Fig 4 on the Time-axis, and is the y-scale in Fig 5 a rate constant measurement?

[Response] Regarding the images in Fig. 4B on the time-axis, another reviewer suggested to delete them, so we decided to delete them in the revised manuscript.

Regarding to the scale in Fig. 5, yes, that is a rate constant. To make this clear and avoid reader’s confusion, the detailed description was made in the figure legend of Fig. 5 in the revised manuscript.

R3-comment_3:  Additionally, what is the scale for the colored bar on Fig 3? Intensity? The area where the brain in located should also be highlighted in the figure or figure legend where the brain is anticipated to be located to further emphasize and differentiate the differences in the distribution of both nitroxides. This may not be apparent to readers.

[Response] Regarding the colored bar on Fig. 3, yes, that was “intensity”.

Regarding emphasizing the brain area, we can add the arrow or circle to indicate the brain region. However, this paper is Review article, therefore, we did not do both descriptions.

To help readers understand this review well, the word “Intensity” was added in the new Fig. 3 of the revised manuscript. Also, the brain area was indicated by the circle in the new Fig. 3 of the revised manuscript.

R3-comment_4: The authors comment that nitroxide probes distribution and kinetics in vivo has not been well described. While this may be true for several of the nitroxides discussed in the manuscript, a review of the literature will reveals several manuscripts which discuss these parameters, particularly with reference to EPR imaging applications, of very similar nitroxides in vivo, in the brain.

[Response]  In this review, we emphasized to develop a new type of EPR imaging system which make it possible to do fast data acquisition and co-registration with anatomical image of an object taken by MRI. With this system, it is possible to visualize detailed distribution and kinetics of nitroxide imagine probes within mouse brains, compared to previous works. Previous reports did not show site-specific reduction rates of nitroxide probes within brains accurately. However, with our system, it is possible to distinguish nitroxide reduction rates within cerebral cortex, midbrain, cerebellum, and others, site-specifically.

Based on these results, we added additional discussion at the last part of the revised manuscript. Added sentences were shown below.

From line 314 in original manuscript: “Conventional EPR imaging system did not show site-specific redox information in rodent brains. However, with new EPR imaging system, detailed information of nitroxide distribution and kinetics within rodent brains can be obtained more acculately.”

R3-comment_5:  The authors should describe or distinguished what is meant by 3D EPR imaging compared 2D-spatial mapping. Are these interchangeable or are the key differences in methodology. 

[Response]   In our studies, 3D EPR images of mouse heads were obtained using projection data sets and the filtered-back projection algorithm. The image matrix of examined mouse heads was 128×128×128. The slice-selective images such as coronal, sagittal, and horizontal images (matrix, 128×128) , were generated from the reconstructed 3D image data sets of mouse heads (128×128×128). Therefore, 2D spatial images were obtained from 3D images. “2D mapping” is the method to map the intensity at each pixel in two-dimensions.  

The detailed description of these words is beyond the scope of this review, therefore, the important reference was added to the reference list of the revised manuscript.

R3-comment_6:  Lines 142-144 are confusing as written. 

[Response]   The following sentence was used in the revised manuscript.

“ Low-molecular weight, stable nitroxide free radicals have been used as reporter probes in spin-labeling biochemistry and biophysics, with an X-band EPR spectrometer is employed to detect such information.”

Reviewer 4 Report

This paper reviews recent developments in the brain redox imaging using in vivo electron paramagnetic resonance imaging and a nitroxide imaging probe. Reactive oxygen species (ROS) and redox alterations contribute to multiple brain conditions and non-invasive assessment of brain redox status is necessary for studies of these pathological conditions. Authors describe basic principles for continuous-wave electron paramagnetic resonance (CW-EPR) imaging using redox-sensitive imaging probe, nitroxides. This is a potentially interesting work but requires some improvements.

1) Author state that “nitroxide-based imaging probes have not been well described for their distribution and kinetics in vivo”, however, they do not offer any solutions. Additional discussion is required.

2) Authors should correct some statements such as “the oxidative damage of ROS” to more biologically meaningful “ROS-mediated oxidative damage” or simply “oxidative damage”.

3) The paper can benefit from discussion of recent EPR imaging studies in other relevant fields such as cancer, for example, Antioxid Redox Signal. 2018;28(15):1365-1377.  

4) Please note that many nitroxide probes used in this paper are not novel. So, the statement “Novel piperidine nitroxides such as Tempol” is not correct. This paper will benefit from inclusion of recent literature of really novel nitroxides available for EPR imaging such as described in Molecules. 2018 Apr 27;23(5). pii: E1034.

5) Authors may want to clarify the specific chemistry involved in the “redox imaging” incorporating to text or Figure a chemical reaction such as

a. TEMPOL     +   1e- àTEMPOL-H

b. TEMPOL-H + ROS à TEMPOL

Author Response

Reviewer-4 (R4)

Thank you very much for reviewing our manuscript and your interest in our research.

R4-comment_1: Author state that “nitroxide-based imaging probes have not been well described for their distribution and kinetics in vivo”, however, they do not offer any solutions. Additional discussion is required.  

[Response]  We realized that this sentence mislead readers and was not appropriate. The sentence “nitroxide-based imaging probes have not been well described for their distribution and kinetics in vivo” appeared at the abstract part, in the original manuscript, and firstly this sentence was deleted, then a new sentence was added in the revised manuscript, as shown below.  

At abstract part, this sentence was deleted, and lines 22-25 were modified, as shown below.

“In this review we discuss the instrumentation, co-registration of EPR image to anatomical images, and appropriate nitroxide imaging probes, all of which are needed for the sophisticated in vivo EPR imager for all rodents. Using new EPR imaging system, site-specific distribution and kinetics of nitroxide imaging probes in rodent brains can be obtained more accurately, compared to previous system.”

R4-comment_2: Authors should correct some statements such as “the oxidative damage of ROS” to more biologically meaningful “ROS-mediated oxidative damage” or simply “oxidative damage”.  

[Response]  According to reviewer’s suggestion, “oxidative damage” was used in the revised manuscript, instead of “oxidative damage of ROS”. 

R4-comment_3: The paper can benefit from discussion of recent EPR imaging studies in other relevant fields such as cancer, for example, Antioxid Redox Signal. 2018;28(15):1365-1377.   

[Response]  Thank you very much for a good suggestion. We always enjoy his work. According to your suggestion, Dr. Khramtsov’s paper was included in the revised manuscript.

1.         Khramtsov, V.V. In Vivo Molecular Electron Paramagnetic Resonance-Based Spectroscopy and Imaging of Tumor Microenvironment and Redox Using Functional Paramagnetic Probes. Antioxid Redox Signal 2018, 28, 1365-1377, doi:10.1089/ars.2017.7329.

R4-comment_4: Please note that many nitroxide probes used in this paper are not novel. So, the statement “Novel piperidine nitroxides such as Tempol” is not correct. This paper will benefit from inclusion of recent literature of really novel nitroxides available for EPR imaging such as described in Molecules. 2018 Apr 27;23(5). pii: E1034. 

[Response]  According to reviewer’s suggestion, the word “Novel” was deleted in the revised manuscript. Also, from reviewer’s suggestion, the paper in Molecules was added to the refence list of the revised manuscript.

R4-comment_5:  Authors may want to clarify the specific chemistry involved in the “redox imaging” incorporating to text or Figure a chemical reaction such as

a. TEMPOL  +   1e- àTEMPOL-H

b. TEMPOL-H + ROS à TEMPOL

[Response]  The reviewer’s text was garbled, shown above, but my understanding is that the reviewer advised us to write the following chemical reaction of nitroxides.

If this is correct, the following chemical reactions were included in the text of the revised manuscript.
